# Hyperglycemia in Pregnancy and Women’s Health in the 21st Century

**DOI:** 10.3390/ijerph192416827

**Published:** 2022-12-15

**Authors:** Harold David McIntyre, Jens Fuglsang, Ulla Kampmann, Sine Knorr, Per Ovesen

**Affiliations:** 1Mater Research, The University of Queensland, South Brisbane, QLD 4101, Australia; 2Department of Clinical Medicine, Aarhus University, 8000 Aarhus, Denmark; 3Steno Diabetes Center Aarhus, 8200 Aarhus, Denmark; 4Department of Obstetrics and Gynaecology, Aarhus University Hospital, 8200 Aarhus, Denmark

**Keywords:** pregnancy, gestational diabetes, obesity, life course perspective, hypertension in pregnancy

## Abstract

Hyperglycemia is the commonest medical condition affecting pregnancy and its incidence is increasing globally in parallel with the twin epidemics of diabetes and obesity. Both pre-pregnancy diabetes and gestational diabetes are associated with short term pregnancy complications, with the risk of immediate complications generally broadly rising with more severe hyperglycemia. In this article we firstly consider these risks and their optimal management during pregnancy and then broaden our scope to consider the long-term implications of hyperglycemia in pregnancy as it relates to overall maternal and offspring health in a life course perspective.

## 1. Introduction

Hyperglycemia in pregnancy is the most common medical complication of pregnancy encountered across all countries and settings. Its prevalence is known to be increasing, in parallel with the dual global epidemics of obesity and diabetes, accompanied by widespread societal changes which favor childbearing at a later age [1].

For the purposes of this paper, we shall use the definition and classification structure proposed by the International Federation of Obstetrics and Gynaecology (FIGO) in 2015 [2], which designates “Hyperglycemia in pregnancy” (HIP) as an umbrella term covering all degrees of severity of hyperglycemia. This broad category is then separated into (overt) “Diabetes in pregnancy” (DIP) (characterized by glucose levels consistent with diabetes, diagnosed either before or during the index pregnancy) and “Gestational diabetes” (GDM) which comprises a large majority of HIP cases [3] and is characterized by levels of hyperglycemia below the thresholds for DIP, diagnosed for the first time during pregnancy. Although this is a practical framework for defining various categories of hyperglycemia, previous publications from a diverse range of countries have used different definitions and descriptive terms, so at times we shall also use the term “Pre-pregnancy diabetes” to describe those cases in which diabetes is known to have been present prior to pregnancy. This category includes Type 1 diabetes (T1D), Type 2 diabetes (T2D) and other specific subtypes including monogenic diabetes (MODY). Currently, the glucose thresholds used to define overt diabetes are largely accepted across various countries and by a range of professional bodies. However, the diagnostic approach to GDM and the precise glucose thresholds used to define it remain controversial, with wide variations between various regions and entities.

As the prevalence of diagnosed and undiagnosed diabetes and prediabetes increases in many countries, what we commonly term “gestational diabetes”, is, in many cases, not truly a “gestational” condition, but rather represents pre-existing impaired glucose metabolism which has remained undetected prior to routine testing during pregnancy. For example, in the USA, surveys outside pregnancy show that diabetes affects 0.6% and prediabetes 13.2% of females aged 12–19 years, whilst by age 20–44 years, diabetes affects 4.5% and prediabetes 29.3% of the population [4,5]. Despite these findings, a GDM prevalence of 17–26% across the USA based centers in the Hyperglycemia and adverse pregnancy outcomes study (HAPO) [6] is considered excessive by some authors. Thus, there appears to be a “reality gap” between the demonstrated population prevalence of impaired glucose metabolism and what is considered worthy of detection and treatment during pregnancy.

The purpose of the current manuscript is to outline temporal tends and the current status of HIP as a contributing factor towards both immediate pregnancy complications and longer-term health of both mother and infant, over a life course perspective. A graphical summary is provided in Figure 1.

## 2. Hyperglycemia in Pregnancy and Immediate Pregnancy Outcomes

Hyperglycemia in pregnancy (HIP) can be classified as either preexisting diabetes, diabetes in pregnancy (DIP) or gestational diabetes mellitus (GDM). in the non-pregnant state. Most cases of HIP are due to GDM (80.3%). While 10.6% are preexisting diabetes and 9.1% are due to overt diabetes diagnosed during pregnancy as depicted in Figure 2 [7,8].

GDM is generally diagnosed by glucose criteria alone. As noted above, it may represent undiagnosed pre pregnancy impaired glucose metabolism. In those women who are considered normal pre pregnancy, GDM arises when insulin secretory capacity is insufficient to overcome the diminished action of insulin (insulin resistance) due to hormone production by the placenta as pregnancy progresses [9]. There is no global consensus regarding the optimal process and threshold values to identify and diagnose pregnant women with GDM. However, most countries and societies agree that the diagnosis of GDM should be made with a 75 g oral glucose tolerance test (OGTT) performed between 24 and 28 weeks of pregnancy. Most countries recommend that all women should be screened/tested while other countries only test women with risk factors for GDM. Maternal overweight and obesity, maternal age, previous history of GDM, family history of type 2 diabetes and ethnicity are GDM risk factors [8].

Recent research has also highlighted the important contribution of genetic factors and gene-environment interactions to GDM risk and has noted that the heterogeneity of definitions and underlying pathophysiology make accurate determination of phenotype-genotype interactions more challenging [10]. One recent publication has identified that all individual values of the 2 h diagnostic OGTT are associated with an overall genetic risk score calibrated for both fasting venous plasma glucose and risk of T2D [11]. Recent evidence from genome-wide association analysis confirms the common genetic links of GDM and T2D, but has also identified some genetic determinants specific to glucose metabolism in pregnancy [12]. Although these studies provide valuable insights into the pathogenesis of GDM, the current high cost of genetic testing precludes their broad implementation into routine clinic practice.

Furthermore, there is no agreement on which glucose levels should be considered diagnostic of GDM in the OGTT. A recent publication from New Zealand [13] has further challenged the glucose levels used for the diagnosis of GDM, comparing higher glycemic criteria for GDM diagnosis with lower criteria in a randomized study. The criteria for the low glycemic group were fasting venous plasma glucose (FVPG) level of 5.1 mmol/L (92 mg/dL) and/or a 2 h level of 8.5 mmol/L(153 mg/dL). For the high glycemic group, the criteria were a FVPG level of 5.5 mmol/L (99 mg/dL) and/or a 2 h level of 9.0 mmol/L (162 mg/dL). The primary outcome was the proportion of infants born large for gestational age (LGA) and this did not differ between groups. Multiple measures of maternal health and infant health were considered as secondary outcomes. GDM was diagnosed in 15.3% of the low glycemic group and in 6.1% of the high glycemic group. The study concluded that use of the lower glycemic threshold led to higher percentage of women receiving a diagnosis of GDM, which led to greater use of healthcare services. This was not, at a population level, associated with a lower risk of adverse events or LGA infants. However, for those women with milder gestational diabetes (treated in the low glycemia group vs. untreated in the high glycemic group), there were health benefits including less preeclampsia, fewer LGA infants and fewer episodes of shoulder dystocia [13].

The lack of international consensus on diagnostic criteria for GDM creates major challenges regarding its true incidence and complicates comparisons between different countries. It results in a wide spread of reported incidence of GDM in different countries-between 1% and more than 30%. However, all reports agree that the rate of GDM is increasing worldwide.

In Denmark, the rate of GDM has increased every year since 2004, despite a consistent approach to diagnosis. In 2021, the rate was 5.9%, an increase of 228% from 2004, with an average increase of 13% per year (Figure 2). 

This is in line with new data from the United States. Among women giving birth in 2020, the overall rate of GDM was 7.8%, an increase of 30% from 2016 [14]. As in the Danish data, the GDM rate rose with increasing pre-pregnancy BMI (Figure 3). Similar tends have been noted in Australia, where the incidence of GDM has risen from 5.2% in 2010 to 16.1% in 2018 [15].

Women diagnosed with GDM have an increased risk of complications of pregnancy and delivery. GDM is associated with increased risk of preterm delivery, pregnancy induced hypertension, caesarean delivery, infants born large for gestational age, admission to a neonatal intensive care unit, neonatal hypoglycemia, hyperbilirubinemia, shoulder dystocia, birth trauma and perinatal death. These risks increase in a linear fashion without a threshold [16,17,18]. A recent systematic review and meta-analysis quantified the short-term outcomes in pregnancies complicated by gestational diabetes mellitus [19]. 

GDM and preeclampsia share many risk factors, especially pre-pregnancy obesity. Barquiel et al. showed that in women with GDM, pre-pregnancy weight status was the strongest risk factor for preeclampsia. Gestational weight gain and glycemic control were also independent risk factors [20]. Similar results were found in the HAPO study [21].

Macrosomia, variably defined as a birth weight of 4000 or 4500 g, is the most frequently reported fetal complication in GDM. A better expression for detection of excess fetal growth is the term large for gestational age (LGA) or birth weight standard deviation score (Z-score), taking gestational age (GA) (and in some models, fetal sex and ethnicity) into account. This allows premature newborns with excessive fetal growth to be identified. The Pedersen hypothesis first proposed that fetal overgrowth was related to increased transplacental transfer of glucose (substrate), stimulating the release of fetal insulin (growth factor) and subsequently, this additional glucose is stored as body fat in the fetus, resulting in excess fetal growth and adiposity, manifested as macrosomia/LGA [22]. This theory has been further developed by Catalano and Hauguel-de Mouzon. In particular, these authors noted that the mother’s serum lipid level and lipid transfer to the fetus is increased in pregnant women with diabetes [23]. The link between maternal hyperglycemia and neonatal outcomes is well described. As described in the HAPO study, there is a linear relationship between the mother’s glucose level and birth weight as well as the newborn’s insulin level and body fat percentage [16,24].

Most women with GDM are able to control their blood glucose levels through a healthy diet, weight management, exercise and blood glucose monitoring [25]. However, some women need medication, most often insulin and less often metformin to manage hyperglycemia. Two large randomized controlled trials have clearly shown that treatment of GDM improves pregnancy outcomes. Overall, detection and treatment of GDM reduces the risk of serious perinatal complications by 67%, macrosomia by 50%, shoulder dystocia by more than 50%, caesarean section by 20% and the risk of pre-eclampsia by 30–50% [26,27].

If maternal blood glucose levels have been well controlled during pregnancy and there are no concerns about the health of the mother or fetus, awaiting natural onset of labour is reasonable. However, induction of labour will often be offered if spontaneous onset of labour has not occurred by 40 weeks in insulin treated women and by 41 weeks in diet treated women. Earlier delivery may be recommended, especially if ultrasound surveillance shows that the fetus is macrosomic (≥4000 g), or if blood glucose levels have not been well controlled.

A broad range of immediate neonatal complications, including congenital anomalies, neonatal hypoglycemia, respiratory distress, need for admission to neonatal intensive care are all increased in babies born to mothers with hyperglycemia in pregnancy [28]. However, these will not be addressed any further in our review.

## 3. Pre-Pregnancy Care to Improve Pregnancy Outcomes

Women with pre-pregnancy diabetes carry an even higher risk of developing adverse pregnancy outcomes than women with GDM [29]. It is well established that the risks of all of these complications can be reduced if optimal glycemic control is achieved already prior to pregnancy and is maintained during pregnancy [30]. A study by Skajaa et al. [31], including 380 women with T1D with a total of 530 pregnancies considered how pre-pregnancy HbA1c affected the course of HbA1c throughout pregnancy. The authors found that an elevated pre-pregnancy HbA1c was a predictor for poor glycemic control throughout pregnancy. This is to be expected, as the time for obtaining strict glycemic control is limited to a period of less than nine months. They also reported that a very poor pre-pregnancy HbA1c was associated with shorter duration of pregnancy and lower birthweight, probably due to a poor foundation for establishing a well-functioning placenta [31]. To facilitate the best intrauterine conditions for the fetus, women with pre-pregnancy diabetes should therefore be supported to optimize glycemic control both prior to and during pregnancy and establishment of pre-pregnancy counseling should be prioritized. A recommendation for systematic preconception counselling is clearly stated in the guidelines regarding management of diabetes in pregnancy from the American Diabetes Association (ADA) [32], but as pointed out by Murphy et al. in an editorial [33], pre-pregnancy clinics favor well-educated women and the most socioeconomically advantaged women with the lowest risk of adverse pregnancy outcomes. Women at higher risk are much less likely to seek pre-pregnancy care. Thus, if overall pregnancy outcomes are to be improved these inequalities in pre-pregnancy care need to be addressed.

In response to this challenge, Murphy and colleagues developed and evaluated a community-based pre-pregnancy care program (PPC) in the UK with the aim of improving pregnancy preparation in women with known diabetes. In total 842 pregnant women with diabetes were included, of whom 502 attended the study before and 340 during/after the PPC. During/after the PPC, pregnant women with T2D were more likely to achieve target HbA1c (≤48 mmol/L [6.5%]) prior to pregnancy and to take folic acid daily. Women with T1D were referred to antenatal care earlier after the PPC. PPC thus contributed to an improvement in pregnancy preparation in both women with T1D and T2D [34].

Aiming to obtain optimal glycemic control both prior to pregnancy and during pregnancy, Feig et al. performed the CONCEPTT study [35], which showed that using continuous glucose monitoring (CGM) compared to capillary glucose monitoring in women with T1D resulted in an improvement in HbA1c during pregnancy. Neonatal health outcomes were significantly improved with lower incidence of LGA, fewer neonatal intensive care admissions and fewer incidences of neonatal hypoglycemia. However, CGM was not superior to capillary glucose monitoring in the women with T1D who were planning pregnancy.

## 4. Early Postpartum Follow Up and Integration of Early Follow Up into Preconception Care

Most studies have evaluated glycemic control before and during pregnancy in women with pre-pregnancy diabetes, but consideration of glycemic regulation postpartum is also important. In particular, glycemic control frequently deteriorates shortly after delivery. Accordingly, Cyganek et al. [36] followed 254 women with T1D and evaluated their glycemic control and weight change after a singleton pregnancy. The study showed that women with T1D had a deterioration in their glycemic control and were unable to return to their pre-pregnancy weight one year postpartum. This was confirmed in a retrospective study by Riskin-Mashiah et al. [37] who evaluated 166 women with T1D one year postpartum. Here, the authors found that there was a rapid deterioration of glycemic control, reinforcing that women with pre-pregnancy diabetes require special attention after delivery to maintain good glycemic control and to return to their pre-gestational weight. Additionally, worth noting is the observation by Skajaa et al., who found that parity affects insulin requirements and interestingly suggested that women with T1D who are planning another pregnancy should be informed that they most likely will need more insulin in the next pregnancy as insulin resistance worsens with rising parity [38].

## 5. Postpartum Follow Up after GDM

Women with GDM have an increased risk of long-term complications, including impaired glucose metabolism, diabetes, cardiovascular disease, and obesity. Catalano et al. [39] have shown that there is a uniform decrease of 50–60% in insulin sensitivity with advancing gestation in both women with normal glucose tolerance and in women with GDM. Women with GDM have lower insulin sensitivity in late pregnancy, compared to women with normal glucose tolerance. This is primarily a reflection of decreased insulin sensitivity that exists prior to pregnancy. Therefore, it is important to have a follow up program for women with previous GDM.

Clinical guidelines recommend postpartum (PP) re-evaluation for prediabetes and diabetes in women with GDM, but vary in their specific protocols. The ADA [32] recommends a 75 g oral glucose tolerance test (OGTT) using non-pregnancy diagnostic criteria to identify prediabetes or diabetes at 4–12 weeks PP, while National Institute for Health and Care Excellence (NICE) [40] recommends fasting plasma glucose (FPG) at 6–12 weeks or an HbA1c test after 13 weeks PP. In addition, ADA recommends pp follow-up at 1–3 years with any recommended glycemic test (HbA1c, FPG or OGTT), whereas NICE recommends annual HbA1c testing [40].

Thus, the international recommendations are divergent. The ADA argues that HbA1c is lowered by the increased red blood cell turnover related to pregnancy, by blood loss at delivery, or by the preceding 3-month glucose profile. The OGTT is therefore more sensitive at detecting glucose intolerance up to 12 weeks after pregnancy. In Denmark endocrinologists and obstetricians also recommend an OGTT 3 months PP, but general practitioners commonly follow the NICE guidelines and perform an HbA1c 12 weeks PP. In Australia, early postnatal follow up with an OGTT at 6–12 weeks PP is recommended, with later assessment primarily using HbA1c to determine long term progression towards diabetes. However, in practice, follow up rates remain poor [41] and clinicians are often uncertain regarding best practice recommendations [42]. Quansah et al. [43] recently reported results from a study where 967 women with GDM from 2011 to 2020 had a 75 g OGTT and an HbA1c performed 4–12 weeks PP and FPG and HbA1c were measured 1 and 3 years after pregnancy. The authors found that the prevalence of glucose intolerance was higher using FPG and HbA1c 4–12 weeks PP, compared to an OGTT, but all diagnostic tests had a very low sensitivity to predict glucose intolerance 10 months PP. They considered that a combination of FPG and HbA1c 1 year PP would be a pragmatic and reliable choice to predict future glucose intolerance [43].

It seems difficult to reach a consensus on postpartum testing, but it is indisputable that the risk of developing diabetes and obesity after GDM can be reduced. It is however necessary to develop a program to produce sustainable behavioral change in women with prior GDM and their families, as partners of women with GDM also have an increased risk of developing diabetes [44]. Several ongoing studies are further investigating this issue, including the Face it Study from Denmark [45].

## 6. The Influence of Breastfeeding on Glucose Metabolism

An important, low-cost approach to reducing diabetes risk after GDM is promotion of breastfeeding. Breastfeeding has long been recommended by global health organizations [46] and is not only beneficial for the children but is associated with a reduced risk of T2D and breast cancer [47,48] and promotes maternal weight loss [49].Both the intensity and the duration of breastfeeding affect the risk of developing T2D after GDM. Gunderson et al. [50] found that a higher intensity of lactation was associated with improved fasting glucose and lower insulin levels at 6–9 weeks pp, and Ley et al. [51], in a follow up study including 4372 women with a history of GDM, reported that longer duration of lactation was associated with a lower risk of T2D and lower HbA1c, FPG and C-peptide.

Lactation also has numerous beneficial effects for the child, including weight control [52]. There are, however, numerous obstacles to breastfeeding, especially breastfeeding for longer periods of time. Many of these obstacles relate to individual issues, to concurrent disease, or to personal issues, as well as the newborn being unwilling or unable to commence breastfeeding, for example, due to preterm delivery.

Further, external factors and societal obstacles may have a profound impact on the possibilities for the new mother to commence and maintain lactation. Social policies such as availability of financial support, also have an impact. In cases where maternity leave is self-funded, prolonged breastfeeding will be more difficult. Access to postpartum care for mothers, including early breastfeeding initiatives, differs markedly across the world [53]. Even in high income countries, focus has now been directed to the health effects that might be obtained with better provision for maternal postpartum care and maternity leave [54]. If resources permit, governmental support during maternity leave may provide a way to set uniform standards and ensure uniform opportunities for all mothers, thereby improving health across subsequent generations.

Breastfeeding is also important in women with T1D, but concerns have been raised as mothers with T1D may experience hypoglycemia shortly after breastfeeding [55,56], which can potentially also result in a reduction in insulin dose and subsequently ketoacidosis. However, Ringholm et al. [57] evaluated the risk of night-time hypoglycemia in breastfeeding mothers with T1D and found that this risk was low and similar to a control group that included women with T1D who had not given birth or breastfed in the previous year. In a recent review, Ringholm et al. [58] also concluded that diabetes management in breastfeeding women with T1D should use the same target glucose values as in other non-pregnant women with diabetes. In addition, they recommended sufficient carbohydrate intake and close attention to a likely need for reduction in insulin doses [58].

## 7. Contraception

In general, recommendations regarding contraceptive methods in women with a history of hyperglycemia in pregnancy do not differ from usual recommendations for contraception. The outcomes of the recent pregnancy may influence the method of choice, including a history of delivery by cesarean section, the length of the breastfeeding, and any peri- or postpartum complications. Age and maternal BMI may also come into consideration when choosing a contraceptive method. An intrauterine device containing progesterone is often a useful choice in hyperglycemia or diabetes. Different contraceptive methods may be recommended according to the geographical setting, and in low- or middle-income countries the availability and cost of contraceptive methods may alter preferences as compared to high income countries.

## 8. Long Term Implications of Hyperglycemia in Pregnancy

Increasing awareness of the later health consequences after a complicated pregnancy has led to the concept of pregnancy as a physiological ‘stress-test’ [59]. Thus, adverse outcomes in pregnancy are associated with later adverse health outcomes. This is particularly true for women with hyperglycemia during their pregnancy.

## 9. Pre-Pregnancy Diabetes and Future Health

Undoubtedly, women with pre-pregnancy diabetes who achieve pregnancy overall belong to a healthier sub-group of individuals with DM. Women with severe diabetes-related micro- or macro-vascular complications and co-morbidities may have been advised to avoid pregnancy. For women with known pre-pregnancy diabetes this is illustrated in the study by Knorr et al. where the overall mortality rate for mothers with T1D was 3.4 times increased compared to controls. However, for those achieving an HbA1c < 64 mmol/mol (8.0%) in the first trimester and a normal pre-pregnancy urinary albumin excretion, the mortality rate was comparable to control women.

With established diabetes, advancing age is associated with an increased risk of late complications of DM such as retinopathy, nephropathy, skin ulcers, peripheral neuropathy, and CVD [60]. Usually, most women with pre-pregnancy DM are aware of the associated risks and in higher income countries will be under regular specialist supervision. However, such care is unlikely to be widely available in low/middle income countries and even in higher income countries, access to health care may be limited by geographic isolation or socioeconomic deprivation [61].

## 10. Impaired Glucose Tolerance and GDM

Individuals with a history of GDM are at high risk for later T2D and its associated morbidity and mortality. Compared to women with normoglycemic pregnancy, their risk for later T2D appears to be increased 10-fold [8,62,63,64]. A family history of T2D is associated with later T2D, as is increasing BMI. In women with GDM other features detected during pregnancy, such as early gestational age at diagnosis, high glucose levels at diagnosis, and need for insulin treatment during pregnancy all carry an increased risk for later T2D [8].

Weight retention after pregnancy is a common occurrence, and in women with impaired glucose tolerance and GDM may push them further towards the development of T2D. It has been proposed that pregnancy itself may aggravate progression towards T2D [65], but at one year post-partum women who returned to their pre-pregnancy weight did not appear to have a worse metabolic profile compared to a group of women without a history of pregnancy [66].

## 11. Repeat Pregnancies and Multiple Pregnancies

It has been suggested that the pancreatic beta cells may become “exhausted” after repeat pregnancies. In theory, a longer total duration of exposure to beta cell stress related to the presence of placental hormone secretion and accompanying maternal insulin resistance could have detrimental effects on long term maternal glucose metabolism, including islet cell production and secretion of insulin. This would be predicted to cause worsening glycemic status in repeat pregnancies.

Only sparse literature exists to determine whether this mechanism truly exists and contributes to longer term complications. Using insulin requirements during pregnancy as a surrogate marker for insulin resistance, our group has previously demonstrated that increased parity is associated with higher insulin requirements in T1D women, persisting after adjusting for maternal size [38]. The question of whether having multiple pregnancies increases a woman’s risk of progression to later T1D or T2D remains unanswered. A study in BMI-matched individuals without diabetes and using a Botnia clamp to investigate insulin sensitivity and beta cell function failed to detect any difference between women with 1 or 2 previous deliveries compared to women with 4 or more previous deliveries [67].

Applying a similar dose–response hypothesis, twin pregnancies would be predicted to exert a larger influence on later maternal glycemic control [68]. However, this theory is not currently supported by objective evidence.

## 12. Cardiovascular Risk

When considering pregnancy complications and their association with later adverse health effects, it remains unclear whether the pregnancy complication per se actually causes later adverse health effects or whether both are seen in individuals predisposed by some common underlying etiological factor for the two related complications. As in the case of GDM, hypertensive disorders of pregnancy (gestational hypertension, preeclampsia, HELLP-syndrome) are also associated with later disease. Several studies have demonstrated that later cardiovascular conditions, including arterial hypertension, subclinical coronary artery disease, ischemic heart disease, stroke, and venous thromboembolism (VTE) [69,70,71], are associated with hypertensive disease in pregnancy.

The high risk of developing T2D after a pregnancy with GDM appears to be associated with the individual’s underlying predisposition towards T2D. Some mechanisms, though, appear to connect hyperglycemia in pregnancy and hypertensive disorders of pregnancy. Thus, GDM is associated with later CVD [72,73,74], including hypertension, and hypertensive disorders during pregnancy are associated with later T2D [75] and with long-term weight gain following pregnancy [76]. It should be noted that pregnancy-related hypertension and hyperglycemia share common risk factors, including advanced maternal age and overweight/obesity. Furthermore, several studies, summarized in a recent systematic review [28] have demonstrated the increased risk of preeclampsia in pregnancies with diabetes.

## 13. Other Adverse Pregnancy Outcomes and Later Health Risks

Interestingly, other pregnancy complications, in addition to hyperglycemia and hypertension in pregnancy, are also associated with later disease. Women who experience pregnancy losses, placental abruption, preterm delivery, and deliver SGA neonates are also at risk of later T2D and/or hypertension and cardiovascular disease [77,78]. Undoubtedly, all of these conditions may share common aetiologic factors and therefore may occur concurrently. Pre-pregnancy diabetes and GDM are both associated with an increased risk of preeclampsia. Both T2D and GDM are associated with overweight, again associated with hypertensive disorders of pregnancy. Hypertensive disorders are associated with placental dysfunction, fetal growth restriction, and SGA, or, in the most severe cases, iatrogenic preterm delivery due to acutely compromised fetal well-being. Even spontaneous preterm birth is associated with later CVD [79]. Adverse pregnancy outcomes may therefore be conceived as important, but non-specific markers for later adverse maternal health effects.

Usually, it is not possibly to quantify the individual contributions of hyperglycemia, hypertension and other adverse conditions in pregnancy for later health risks. Since these conditions during pregnancy share common features, may co-exist, and also have some overlap in later health consequences, it seems reasonable to consider them as a cluster of abnormalities requiring increased attention after childbirth. It would be valuable to develop comprehensive postpartum follow-up for women with a history of one or more of these pregnancy complications, aiming at both prevention of progression to overt disease and early intervention.

## 14. Prevention and Early Detection to Reduce Long Term Health Risks

Given the multiple health risks identified above, women with prior GDM should be offered advice and follow-up due to their risk of developing T2D and cardiovascular disease. As these women have been identified at a relatively young age, there is an opportunity for both prevention and early detection of progression towards overt T2D.

Reducing morbidity during subsequent pregnancies should be considered as a key benefit, as well as measures to avoid or diminish later deterioration of insulin sensitivity. The need to focus on breastfeeding, reducing weight retention and promoting weight loss, enhancing physical activity and improving dietary habits is clear. Unfortunately, permanent weight loss has generally proven difficult to achieve without bariatric surgery or pharmacologic interventions. Some have suggested a prophylactic approach with the prescription of insulin sensitizing drugs, principally metformin, to women at an increased risk for later T2DM [80,81]. Women with a history of GDM are clearly an identifiable high-risk group and appear more likely to benefit from metformin than other subgroups, based on evidence from the Diabetes Prevention Program [82]. Thiazolidinediones have also been shown to preserve beta cell function in women with previous GDM, but other adverse events preclude their use in this context [83].

Targeted screening and/or intervention for women with prior GDM may be more feasible compared to a universal screening program. Closer attention to a woman’s pregnancy history by primary care providers provides an opportunity for more focused screening for worsening insulin resistance, T2D, cardiovascular disease, and metabolic syndrome. Unfortunately, many primary caregivers are still unaware of the relationship between pregnancy complications and later health [84]. Given the co-occurrence of T2DM and CVD after complicated pregnancies, including hyperglycemia in pregnancy, follow-up programs should be formulated and directed at multiple risk factors, targeting both T2D and CVD, including hypertension. Many countries do have screening programs to identify T2D and hypertension, often with a focus on both. However, pregnancy complications are generally not included in CVD risk calculators as, e.g., the SCORE-2 [85] or the ASCVD risk estimator Plus [86]. When screening for T2D, a history of GDM may be considered as a component of risk estimates, but other adverse pregnancy outcomes generally do not form part of the risk determination algorithm [87].

In the future, risk assessment for later maternal health should be expanded to include an assessment of a range of adverse pregnancy outcomes. This will then facilitate increased vigilance and a closer focus on the later effects of adverse pregnancy outcomes by health institutions, health authorities, health personnel across specialties, and in the population in general, especially among women of fertile age and their partners. Such a focus is not yet widespread [84,88], and efforts that would disseminate knowledge on the associations between adverse pregnancy outcomes and later health effects merit prioritization.

It is unclear how a follow-up program for women with a pregnancy history of hyperglycemia or hypertension should be optimally designed and implemented. In the “Management of Diabetes in Pregnancy: Standards of Medical Care in Diabetes—2022” from the American Diabetes Association [32], postpartum advice for women with a history of GDM includes regular follow-up visits with 1 to 3 years interval for screening for T2D. The importance of post-pregnancy care has also been recognized by FIGO [89], but recommendations for implementation remain country specific. In case of pre-diabetes, lifelong life-style interventions are recommended and even the prescription of metformin is suggested. Despite evidence from the Diabetes Prevention Program [82], the role for metformin for prevention of later T2DM after GDM is far from universally accepted [81].

In 2021, the American Heart Association published a scientific statement on the risk of later disease after adverse pregnancy outcomes, including GDM and hypertensive disorders of pregnancy. This publication focused on the risk of later disease and recommended that future health care efforts should try to identify individuals with an adverse pregnancy outcome in the early postpartum period in order to institute early prophylactic measures and to commence control programs that might better identify the occurrence of later disease [77]. The optimal design of such a program remains unclear. It has been suggested that screening programs could be combined, for example screening for cervical dysplasia could be combined with screening for hypertension and T2D. National screening programs for follow-up after adverse pregnancy outcomes have, however, not been formulated.

In summary, despite the acknowledgement of higher risks, no clear recommendations for follow-up programs after a pregnancy with adverse outcome exist. Selective rather than universal screening programs for hyperglycemia and/or hypertension may prove to be more efficient. This, in turn, sets new demands for the awareness of health professionals regarding previous obstetric history. A first goal would be to increase awareness among women, their partners, and health care professionals that pregnancy is a “stress-test” and thereby a window for improving a woman’s future health.

## 15. Longer Term Effects of Hyperglycemia in Pregnancy on the Offspring

While reasonable consensus has been established within the area of perinatal mortality and morbidity in pregnancies complicated by HIP, the impact of HIP on the health of the baby, during childhood and adolescence and in adult life is less uniform.

Mortality beyond the perinatal period is very sparsely described. Vääräsmäki et al., in a Finnish study from 2002 [90], reported the total rate of deaths until the age of one year to be higher in offspring born to women with DIP than in non-diabetic mothers (19.9/1000 vs. 8.1/1000). These numbers included stillbirths, early neonatal death (0–6 days), late neonatal death (7–27 days) and post-neonatal death (28–364 days). Only 5 children born to women with DIP died during the post-neonatal period resulting in an OR of 3.8 (1.6–9.2). Additionally, two Danish studies have reported on mortality reaching beyond the perinatal period [91,92]. Only the study by Knorr et al. in which children born to mothers with type 1 diabetes were followed for one year found an increased mortality during their first year of life, compared to children born to mothers from the background population (Hazard Ratio 2.10, 95% CI 1.33–3.30, *p* = 0.001) [91]. By contrast.another study comprising children of both women with DIP and GDM, found no significant difference in mortality for children born to women with HIP compared to unaffected controls (risk difference 0.09%, 95% CI −0.79–0.98) [92].

While no studies have described increased mortality beyond the first year of life in offspring born to women with HIP, the harmful effects of intrauterine hyperglycemia are reflected in both register-based as well as clinical studies of childhood and adolescent health.

A register-based study from the EPICOM cohort, found an increased number of hospital admissions and increased use of medication until the age of 15 years in children born to women with type 1 diabetes [91]. The increase in hospital admissions was associated with maternal pre-pregnancy and first trimester HbA1c. A Swedish register study found comparable increases in hospital admission for neurological/developmental disorders, congenital malformations, infections and accidents, in children of women with DIP and to a lesser degree in children of women with GDM [93].

Several clinical studies have explored the influence of maternal HIP on childhood and adolescent health and most of them describe deleterious effects, increasing with age, on overweight/obesity [94,95,96,97], impaired glucose metabolism [98,99,100,101] and fatty liver disease [102]. Over the last couple of years, studies on sex specific effects of intrauterine exposure to HIP, especially around the time of puberty, have emerged. Several clinical studies have investigated the influence of maternal GDM on pubertal development [95,103,104,105]. Kubo et al. reported that girls born to women with GDM entered puberty earlier than children born to healthy controls [103] and Monteilh et al. describe the same development only in boys [104]. In 2017, Grunnet et al. published data from the Danish Better Health in Generations cohort, showing girls born to women with GDM experiencing earlier puberty when compared to healthy controls [95]. For HIP overall, Hockett et al. described how growth rate, as an expression of onset of puberty, was associated with maternal HIP, especially for girls around 16 years of age [105]. However, inboth the Grunnet and Hockett studies, the association between maternal GDM or HIP and puberty, were attenuated when adjusted for offspring BMI and maternal pre-pregnancy BMI.

Regarding the influence of pre-pregnancy maternal diabetes, only a single study has divided the offspring according to known maternal pre-pregnancy diabetes vs. GDM [106]. The study was based on self-reported information on pubertal development combined with registry data and found no association between maternal pre-pregnancy diabetes and onset of puberty [106]. These authors did, however, find an association between maternal GDM and offspring pubertal development. Childhood overweight probably plays a part in timing of puberty, and especially in girls it is reported that childhood overweight increases the risk of early puberty [107]. Therefore, it is likely that the overweight induced by maternal HIP and/or maternal obesity leads to earlier onset puberty in offspring exposed to maternal HIP in pregnancy.

Studies in adult offspring of women with HIP are sparser. Clausen et al., in 2008 reported an increased risk of T2D or pre-diabetes at the age of 18–27 years and found the risk of developing T2D s or pre-diabetes to be associated with maternal blood glucose in the third trimester [108]. Later the same group described an increased risk of overweight and the metabolic syndrome in the same cohort of offspring born to women with HIP [109]. A Czech study also found higher levels of blood glucose, insulin levels, BMI and both systolic and diastolic blood pressure in offspring born to women who were insulin treated during pregnancy [110].

Sobngwi et al. linked impaired glucose tolerance and a deficient insulin secretory response to glucose, to intrauterine exposure to T1D [111]. However, they did not find any difference in BMI, fat mass (examined with dual X-ray absorptiometry), blood pressure or waist/hip ratio. A large Swedish register-based study explored the association between maternal HIP and offspring risk of adiposity in young men at the time of conscription (n = 1475). Due to limitations in the registry, it was not possible to differentiate between offspring born to women with pre-pregnancy diabetes or GDM. However, they did find an increased risk of adiposity in offspring exposed to HIP compared to non-exposed siblings [112].

In addition, offspring cardiovascular health also seems affected by maternal HIP. Contrary to pubertal timing, the association between cardiovascular disease in the offspring reflects the timing of hyperglycemia in utero, the most severely affected being offspring born to mothers with pre-pregnancy diabetes with lesser effects noted amongst offspring of mothers with GDM [113,114].

However, most of the above-mentioned studies do not consider the effect of paternal body size or glycemic status on later life offspring health. Across the few studies that have examined offspring of mothers with HIP compared to offspring of fathers with diabetes, evidence is not conclusive [111,115,116]. One study found an effect of paternal diabetes on offspring beta cell function [116], while the other two studies used offspring of fathers with diabetes as controls, making comparison difficult. Additionally, for paternal body size, this is known to impact both childhood and adolescent risk of overweight and obesity, and therefore needs to be considered for future long-term studies of offspring born to mothers with HIP [117].

A further area of controversy is the potential association of offspring cognitive and behavioral function with maternal HIP. Many conditions during fetal life, birth complications, neonatal complications and socioeconomic factors contribute to later neuro-cognitive function and differentiation between the contributory factors can be difficult. In Sweden, Dahlquist et al. found that offspring born to women with HIP obtained lower grades when finishing compulsory schooling at the age of 16 and that fewer were able to complete compulsory school [118]. The EPICOM study, considering performance in primary school, found no difference in grades between offspring of mothers with pre-pregnancy type 1 diabetes compared to matched controls, but a negative association between maternal glycaemic control and offspring school performance [119]. On contrary, direct IQ testing of the EPICOM cohort found a slightly, but statistically significant, lower IQ in the children of mothers with pre-pregnancy diabetes and an increased requirement for use of attention deficit hyperactivity disorder (ADHD) medication [120,121]. An increased risk of ADHD has also been described in an American cohort along with an increased risk of an autism diagnosis [122,123]. For both diagnoses, the risk rose with increasing severity of maternal HIP in pregnancy, with children of women with pre-pregnancy type 1 diabetes having the highest risk and children of women with diet treated GDM the lowest risk of these diagnoses.

Three additional studies exploring the Scandinavian registries, have described a negative impact of maternal HIP on offspring cognitive outcomes [124,125]. One examined the school marks at age 16 and IQ assessed at 18 years of age for male offspring of mothers with HIP and found significantly lower grades when finishing compulsory school and a significantly lower IQ at conscription [124]. The study used a sibling design enabling them to study within sibship associations. However, no such association between maternal HIP and offspring IQ was found and this led the authors to conclude that the association between maternal diabetes in pregnancy and offspring cognitive outcome is more likely to be explained by familial characteristics than by intrauterine exposure of T1D.

Another study, supporting the hypothesis of a predominant effect of overall familial characteristics, rather than specific intrauterine exposure to HIP, impacting cognitive impairment in the children, has recently been published. Here, Spangmose and Skipper et al. found both children of mothers and children of fathers with type 1 diabetes, achieved lower test scores than children born to parents without diabetes [125]. Thus, the area of cognitive function and behavioral disturbances following exposure to maternal hyperglycemia in utero, remains controversial and merits further investigation.

## 16. Conclusions

Hyperglycemia is the commonest medical condition complicating pregnancy, despite likely under-ascertainment due to the fact that policies and protocols relating to its detection vary widely across the world [126]. Most cases of HIP are recognized during pregnancy as GDM, and GDM incidence is rising globally in parallel with a rise in the prevalence of overweight and obesity.

HIP has consequences for both pregnant women and their offspring, both during and after pregnancy. Whilst most health care systems attempt to provide care for women with pre-pregnancy diabetes in a “higher–risk” model of care, such care is resource intensive and “best practice” is an aspiration rather than a reality in many situations.

Mothers with HIP are at risk of later disease, first and foremost the development of later type 2 diabetes but also hypertension and other cardiovascular diseases. Their offspring carry increased risks of obesity, impaired glucose metabolism and cardiovascular disease, with possible effects also on neuro-cognitive function with severe maternal hyperglycemia. Unfortunately, little attention is currently paid to improving long term health for either mothers with HIP or their offspring, so an important opportunity to reduce the global burden of non-communicable disease is frequently wasted.

Increased awareness of the consequences of hyperglycaemia in pregnancy in a life-long perspective rather than solely in a pregnancy perspective will increase the possibilities for early prevention and treatment of the long-term health risks for both mother and child associated with hyperglycaemia in pregnancy.

## Figures and Tables

**Figure 1 ijerph-19-16827-f001:**
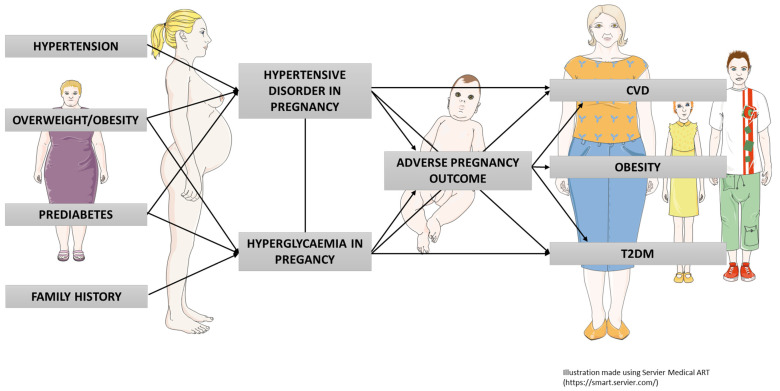
Graphical summary of pregnancy complications and later health implications for mother and child.

**Figure 2 ijerph-19-16827-f002:**
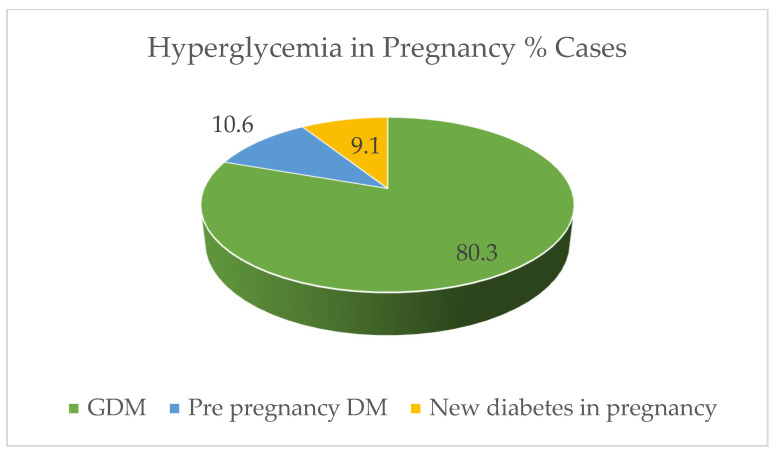
Hyperglycemia in pregnancy. (International Diabetes Federation [7]).

**Figure 3 ijerph-19-16827-f003:**
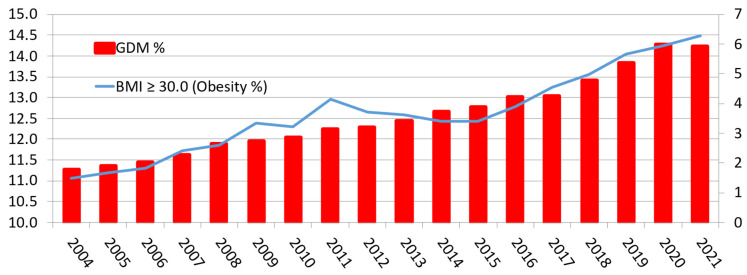
Gestational Diabetes Mellitus (GDM) and BMI ≥ 30 in Denmark 2004–2021 (%) (The Danish Health Data Agency (Sundhedsdatastyrelsen) 2022). “x” axis—Year; Left sided “y” axis—% Pregnant women with BMI ≥ 30 kg/m^2^; Right sided “y” axis—% Pregnant women clinically diagnosed with GDM.

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
