# Peer review of "Hyperglycemia in Pregnancy and Women’s Health in the 21st Century"

_ijerph, 2022, doi:10.3390/ijerph192416827_

Round 1
Reviewer 1 Report
English language edition needed.
Author Response
Thank you for this positive assessment of our work. We note that there are no specific comments regarding revisions. We shall look again at the English language aspects, but we note that this was rated "5 stars" in the summary version, but still noted as requiring revisions. We have carefully reviewed the English spelling and grammar in our revised version
Reviewer 2 Report
The Reviewer finds the proffered work to be largely sound and of significant interest to practitioners and patients alike. To paraphrase, and as Authors astutely note that gestational "....[h]yperglycemia in pregnancy is the most common medical complication encountered across all countries and settings". Breaking out the etiological factors linked to this condition, Authors mention numerous contributory triggers - including age, insulin secretory capacity, HbA1c and postpartum factors such as breastfeeding.
While Authors lightly touch upon 'family history' they fail to address perhaps the most key facet of relevance in familial cohort association - i.e. genetics. Here, and particularly in the past several years, compelling linkage between susceptibility genes and phenotype have been tested and well-documented. To remain relevant in the current field the Authors must consider and address this data shortfall.
Minor typographical misspellings throughout the text predicate a more thorough Author review before resubmission.
Author Response
Thank you for these overall positive comments.
The question of genetics is clearly important. We have now added a paragraph (lines 81 - 91) and 3 additional references, highlighting recent findings related to the genetic risk factors for GDM and including a very recent GWAS study which identifies both factors common to GDM and T2D and some factors specific to pregnancy glucose metabolism. Given the broad scope of our review, we have not included detailed descriptions of specific genetic markers, but these are reading available to readers via the cited references.
Reviewer 3 Report
This REVIEW is very well organized. I only request the author minor changes, such as spelling corrections(Line 70 and 71), addition of units (please add mg/dL to mmol/L) and figure corrections (please explain the axis of Figure2).
Author Response
Thank you for these positive overall comments. In the revised version, we have corrected the spelling errors noted and also clarified the axis labels for Figure 3 (actually incorrectly labelled in the initial submission due to some last minute editing - apologies for this...)
We have also added dual units for both glucose and HbA1c to facilitate understanding for readers from multiple countries